# Unravelling the Role of Habenula Subnuclei on Avoidance Response: Focus on Activation and Neuroinflammation

**DOI:** 10.3390/ijms241310693

**Published:** 2023-06-27

**Authors:** Geiza Fernanda Antunes, Ana Carolina Pinheiro Campos, Daniel de Oliveira Martins, Flavia Venetucci Gouveia, Miguel José Rangel Junior, Rosana Lima Pagano, Raquel Chacon Ruiz Martinez

**Affiliations:** 1Division of Neuroscience, Hospital Sírio-Libanês, Sao Paulo 01308-060, Brazil; 2Neurosciences and Mental Health, The Hospital for Sick Children, Toronto, ON M5G 0A4, Canada; 3Centro Universitário de Santa Fé do Sul, Santa Fé do Sul 15775-000, Brazil; 4Medical School, Universidade Brasil, Fernandópolis 15600-000, Brazil; 5Laboratorios de Investigação Médica—LIM/23, Institute of Psychiatry, School of Medicine, University of Sao Paulo, Sao Paulo 05508-900, Brazil

**Keywords:** avoidance behavior, lateral habenula, medial habenula, astrocytes, neuroinflammation, neuronal activation

## Abstract

Understanding the mechanisms responsible for anxiety disorders is a major challenge. Avoidance behavior is an essential feature of anxiety disorders. The two-way avoidance test is a preclinical model with two distinct subpopulations—the good and poor performers—based on the number of avoidance responses presented during testing. It is believed that the habenula subnuclei could be important for the elaboration of avoidance response with a distinct pattern of activation and neuroinflammation. The present study aimed to shed light on the habenula subnuclei signature in avoidance behavior, evaluating the pattern of neuronal activation using FOS expression and astrocyte density using GFAP immunoreactivity, and comparing control, good and poor performers. Our results showed that good performers had a decrease in FOS immunoreactivity (IR) in the superior part of the medial division of habenula (MHbS) and an increase in the marginal part of the lateral subdivision of lateral habenula (LHbLMg). Poor performers showed an increase in FOS in the basal part of the lateral subdivision of lateral habenula (LHbLB). Considering the astroglial immunoreactivity, the poor performers showed an increase in GFAP-IR in the inferior portion of the medial complex (MHbl), while the good performers showed a decrease in the oval part of the lateral part of the lateral complex (LHbLO) in comparison with the other groups. Taken together, our data suggest that specific subdivisions of the MHb and LHb have different activation patterns and astroglial immunoreactivity in good and poor performers. This study could contribute to understanding the neurobiological mechanisms responsible for anxiety disorders.

## 1. Introduction

Anxiety disorders are the most prevalent mental disorder, with different and complex conditions that significantly reduce the quality of life of affected individuals [1,2]. Due to its complex, heterogeneous and individualized nature, understanding the biological mechanisms that are altered in anxiety disorders is a major challenge. Avoidance behavior is an essential feature of anxiety disorders [3,4]. In preclinical research, avoidance response can be evaluated using the two-way active avoidance test [5,6]. Based on the number of avoidance responses, it is possible to differentiate two populations of animals: the good performers and the poor performers. While poor performers freeze excessively and exhibit less than 20 avoidance responses in a trial, good responders are able to learn the task and avoid the aversive stimulus [6].

It has been proposed that this behavioral distinction could be based on differential recruitment of brain circuits [7,8]. One structure that is implicated in the neuronal network of psychiatric disorders [9] and has not been well-evaluated in this paradigm is the habenula. The habenula is a limbic structure classically involved in motivation, emotion, reinforcement, learning, pain and depression [10,11]. However, it has been emerging as responsible for processing aversive stimuli in an experience-dependent selection of behavioral responses to stressors [12,13]. Based on the pattern of afferent and efferent connectivity, the habenula can be divided into several subnuclei that have distinct cellular and molecular characteristics [14,15,16]. These anatomical and molecular divisions highlight the complexity of the habenula [14,17], emphasizing the importance of a detailed evaluation of this structure. The habenula plays a central role in the connections between the forebrain to midbrain regions for the integration of emotional and sensory processing [18]. It is also considered a major regulatory site for serotonergic and dopaminergic neurotransmission [19,20], both of which are highly implicated in the neurobiology of anxiety disorders and avoidance behavior [21]. In line with this, it has been shown that lesions targeting the LHb facilitate avoidance responses [22], while stimulation impairs this response [23]. However, the effect of aversive learning performance in the different subnuclei of habenula remains unclear.

Another important factor that may influence habenular function is the functional integrity of astrocytes, which play an important role in immune responses, synaptic pruning and neuroplasticity [24]. It has been proposed that chronic stress induces an increased inflammatory response in the habenula [25], suggesting that glial cells could contribute to the local inflammatory profile, resulting in a maladaptive and chronic inflammatory state [26], playing an important role in chronic neuroplasticity [27].

Therefore, this study aimed to perform a broad evaluation of the neuronal activation patterns (FOS expression) and astrocyte density (GFAP immunoreactivity) in several habenula subnuclei in rats presenting good and poor performance and control animals in an aversive learning paradigm to investigate if these patterns could help explain the behavioral difference observed in these groups. Specifically, this study aimed to shed light on the habenula subnuclei signature in avoidance behavior, evaluating the pattern of neuronal activation using FOS expression and astrocyte density using GFAP immunoreactivity, and comparing control, good and poor performers.

## 2. Results

### 2.1. Behavioral Testing: Two-Way Active Avoidance Test

Based on the number of avoidance responses presented during training, animals were divided into good and poor performance groups. The good performers showed an increase in the number of avoidance responses (Group F_(1,17)_ = 66.97, *p* < 0.001; Session: F_(7,17)_ = 5.97, *p* < 0.001; Group × Session F_(7,17)_ = 6.578; *p* = 0.0007; Figure 1A) and a decrease in the percentage of freezing (Group: F_(2,32)_ = 68.82, *p* < 0.001; Session: F_(3,32)_ = 5.36, *p* = 0.002; Group × Session F_(2,32)_ = 77.96; *p* < 0.0001; Figure 1B) along the sessions.

### 2.2. Immunoreactivity in the Medial Habenular Complex: MHbS, MHbI, MHbC and MHbL

The medial habenular complex was evaluated in its subnuclei: MHbS, MHbI, MHbC and MHbL.

#### 2.2.1. Activation Pattern

Animals in the good performance group presented less FOS-IR in the MHbS (F_(2,6)_ = 22.58; *p* = 0.0016—Figure 2A) when compared with control and poor performance groups. The poor performance group showed increased FOS-IR in the MHbL (F_(2,8)_ = 9.27; *p* = 0.0082—Figure 2B) when compared with the control group. There was no statistical difference in FOS-IR in the MHbC (F_(2,8)_ = 1.30; *p* = 0.32) and MHbI (F_(2,8)_ = 2.56; *p* = 0.14) parts.

#### 2.2.2. GFAP—Astroglial Immunoreactivity Pattern

Poor performance animals exhibited an increase in GFAP-IR in the MHbI (F_(2,7)_ = 14.50; *p* = 0.0032—Figure 2C) when compared with control and good performers. There was no statistical difference in the other regions of the MHb, MHbS (F_(2,9)_ = 1.56; *p* = 0.26), MHbC (F_(2,9)_ = 1.79; *p* = 0.23) and MHbL (F_(2,10)_ = 0.03; *p* = 0.96).

### 2.3. Immunoreactivity in the Medial Division of the Lateral Habenula: LHbMS, LHbMPc and LHbMC

The medial division of the lateral habenular complex includes the LHbMS, LHbMPc and LHbMC.

#### 2.3.1. Activation Pattern

In LHbMS, there was an increase in FOS-IR in good performers in comparison with the control group (F_(2,8)_ = 10.87; *p* = 0.0052—Figure 3A). In LHbMPc, the good and poor performers showed an increased FOS-IR in comparison with control animals (F_(2,8)_ = 20.79; *p* = 0.0007—Figure 3B). In LHbMc, there was no difference between the groups (F_(2,8)_ = 1.29; *p* = 0.32).

#### 2.3.2. GFAP—Astroglial Immunoreactivity Pattern

There was no statistical difference in the IR-GFAP staining in LHbMS (F_(2,7)_ = 2.07; *p* = 0.20), LHbMPc (F_(2,8)_ = 1.39; *p* = 0.30) and LHbMC (F_(2,6)_ = 0.90; *p* = 0.45) between groups.

### 2.4. Immunoreactivity to FOS in the lateral division of the lateral habenula: LHbLMc, LHbLO, LHbLB, LHbLPc and LHbLMg

The lateral division of the lateral habenular complex includes the LHbLMc, LHbLO, LHbLB, LHbLPc and LHbLMg.

#### 2.4.1. Activation Pattern

In the LHbLB and LHbLPc, there was an increase in FOS-IR in poor performers compared with all other groups (F_(2,8)_ = 25.20; *p* = 0.0004—Figure 4A; (F_(2,8)_ = 5.19; *p* = 0.034—Figure 4B). In the LHbLMg, the good performers showed an increase in FOS-IR in comparison with poor performers and control animals (F_(2,8)_ = 25.20; *p* = 0.0004—Figure 4C). No statistical difference was observed in the LHbLMc (F_(2,7)_ = 0.026; *p* = 0.97) and LHbLO (F_(2,7)_ = 5.03; *p* = 0.44).

#### 2.4.2. Astroglial Immunoreactivity

The LHbLO subnuclei presented a decrease in the quantification of GFAP-IR in good performers when compared with the control group (F_(2,6)_ = 7.570; *p* = 0.022—Figure 4D). The LHbLB showed a reduction in IR-GFAP in good and poor performers when compared with the control group (F_(2,8)_ = 10.07; *p* = 0.0065—Figure 4E).

There was no statistical difference between groups in the LHbLMc (F_(2,7)_ = 2.77; *p* = 0.13), LHbLPc (F_(2,10)_ = 1.36; *p* = 0.30) and LHbLMg (F_(2,7)_ = 4.51; *p* = 0.055).

Table 1 summarizes the changes in behavior response, FOS-IR and GFAP-IR in good and poor performers.

## 3. Discussion

To our knowledge, our study is the first to investigate the role of habenula subnuclei in the behavioral distinction between good and poor performers in the two-way active avoidance test. Good performers displayed an increased number of avoidance responses and a decrease in the freezing response, whereas poor avoiders showed an opposite pattern [7]. Our data showed that the good performers had a decrease in FOS-IR in the MHbS and an increase in the LHbLMg, while the poor performers showed an increase in FOS-IR in the LHbLB and an increase in GFAP-IR in the MHBI. This work showed that good and poor performers have a distinct pattern of neuronal activation and astroglial reactivity in several subnuclei of the habenula.

The habenula has been shown to be involved in the modulation of avoidance behavior. Initial studies have explored lesions targeted to the habenular complex [28,29], showing impairment in avoidance response, suggesting that its connection with the limbic system is a crucial interface to evaluate the aversive stimulus [30]. Evidence over the years indicates that habenula shares more heterogeneous subdivisions and connections [31,32], highlighting the complexity and importance of the hub of connections that the habenula participates in.

### 3.1. The MHb Activation Pattern and Astroglial Reactivity in Good and Poor Performers in the Two-Way Aversive Learning Paradigm

Without a doubt, the role of MHb in avoidance behavior has been less investigated when compared with LHb. However, it has been shown that lesions in the MHb increased freezing behavior in zebrafish [33]. The MHb is the major habenular connection to the interpeduncular nucleus (IPN) by cholinergic and glutamatergic fibers [34,35]. Moreover, the MHb also projects into the epithalamus and LHb [17]. The IPN is an important structure of the brain that connects the habenular complex and the monoaminergic systems, contributing to the regulation of learning, sleep, reward, executive planning and fear response [14,16,36,37,38]. Furthermore, when considering fear and anxiety, the connection between the MHb and the posterior septum also points to the importance of the MHb in the modulation of these behaviors [33]. In this sense, Klemm [39] suggested that the interaction between the posterior septum, MHb and IPN directly reflects emotional processing and mental disorders that could occur upon failure in this pathway [39]. To better comprehend the effect of aversive learning performance in the MHb, here, we performed a subdivision of the MHb as proposed by Andres and colleagues [31]. We showed that the good performers have a decreased neuronal activation pattern in the MHbS when compared with all the other groups. Interestingly, this is the only division of the MHb that substance P and cholinergic neurons are absent, while noradrenergic fibers and a high expression of IL-18 and GABAergic interneurons can be found [40]. However, the role of noradrenaline as a possible inhibitor on the MHb of good performers has yet to be investigated.

Interestingly, considering that our animals were evaluated after the last session (eighth session) in the shuttle-box, the decreased neuronal activation observed in these nuclei may be related to the lack of activation when the task was already learned, due to the fact that good performers had showed an increase in avoidance since the fourth session. Moreover, we found that poor performers showed an increase in FOS-IR in the MHbL and astroglial immunoreactivity in the MHbI. While both of these subdivisions have substance P and cholinergic co-expressed neurons, only the MHbL shows colocalization between glutamatergic and mu-opioid-receptor, but also P-type calcium channels, neurokinin-3 receptors and the distribution of intensely stained metenkephalin-positive fibers [40,41,42,43]. These areas containing substance P, cholinergic and glutamatergic neurons project slightly exclusively to the IPN [44], suggesting that poor performance could result in the activation of the MHb-IPN glutamatergic-cholinergic pathway. The increase in FOS-IR in poor performers may reflect the increased rate of acute freezing behavior in the last session, which suggests a participation of the MHb-IPN in fear and anxiety [45,46]. Again, further studies are needed to better comprehend the extent of each MHb-IPN projection since different neurons within the MHb have very distinct roles. For example, while the activation of GABAergic receptors in the MHb-IPN pathway induces increased neurotransmitter release in the glutamatergic-cholinergic projection, it has the opposite effect in glutamatergic-substance P projections [47]. On the other hand, the increased astroglial IR showed in poor performers may reflect maladaptive neuroplasticity from consistent stress and freezing behavior due to the failure to learn the task. Increased astroglial IR is linked to increased neuroinflammation induced by stress [48]. However, astrocytes may also increase their ramifications hours after the occurrence of long-term potentiation [49], suggesting that MHbI subnuclei may be more sensitive towards maladaptive neuroplasticity of fear and anxiety consolidation. Nevertheless, our intriguing results suggest that MHb also plays a pivotal regulatory role in avoidance behavior, where each subdivision demonstrates a particular signature.

### 3.2. The LHb Activation Pattern and Astroglial Reactivity in Good and Poor Performers in the Two-Way Aversive Learning Paradigm

The LHb is the major portion of the habenula, with most studies focused on aversive learning. Lesions to the LHb lead to impairment in avoidance responses [22,23]. We found that good and poor performers showed an increase in the neuronal activation pattern of the LHbMPc when compared with control animals. Considering that both good and poor performers are subjected to the avoidance protocol, it is possible to hypothesize that this activation may be related to the stress response rather than learning performance. Notably, animals were evaluated after eight sessions in the shuttle box, suggesting that LHbMPc may be sensitive to stress even after consecutive sessions. Supporting our data, it has been shown that a variety of stressor could increase FOS expression in the lateral habenula, emphasizing the close interaction with the medial prefrontal cortex, lateral septum, extended amygdala, hypothalamus and dorsal raphe [50,51,52,53,54]. In a similar way, both groups showed a decrease in astroglial immunoreactivity in the LHbLB. Decreased astroglial IR has been related to depressive-like behavior in different areas of the brain, but especially the hippocampus [55,56]. Additionally, astroglial activation to stress in the habenula has been shown [57,58]. Because these results were found in good and poor performers, it suggests that the chronic stress-induced disruption of pathways is independent of learning performance. Indeed, it has been shown that the LHb participates in behavioral responses such as pain, anxiety, reward and stress [50,59,60,61]. Although the habenular complex may not be directly involved in regulating the effects of stress on the HPA axis, this structure has been associated with a variety of behaviors that are influenced by stress, including learning, exploratory behavior and responsiveness to aversive stimuli [62,63]. Furthermore, behaviors such as exploratory activity, responsiveness to aversive stimuli and sexual behavior present circadian variations [64] that may, in turn, be related to the activation pattern of the LHb [65].

Here, we conducted a detailed evaluation of the many subdivisions of LHb. We found that poor performers showed an increased neuronal activation pattern in the LHbLB and LHbLPc. These nuclei have distinct cell types (i.e., cholinergic and parvalbumin neurons, respectively [66]), suggesting that poor performance in avoidance behavior involves a dysregulation in the habenular complex. Moreover, both structures receive inputs from the magnocellular preoptic nucleus, which has an important role in neuroendocrine responses [67]. Notably, while good and poor performers showed a decrease in astroglial IR in the LHbLB, only poor performers showed an increase in the neuronal activation pattern. FOS-IR is often attributed to an acute response, suggesting that this subnucleus is involved in learning performance. The reduction in GFAP-IR observed in both groups may be a result of chronic stress induced by the shuttle box testing.

On the other hand, good performers showed an increase in the activation pattern in the LHbLMg and LHbMs. The LHbM nuclei receive important projections from the entopeduncular nucleus, lateral hypothalamus, areas of the limbic system and the ventral striatal complex, such as the prefrontal cortex [14,68,69,70]. Interestingly, the LHbMS, which is more activated in good performers, contains somatostatin, which may facilitate learning and cognitive function [71].

We found that good performers have a decrease in astroglial immunoreactivity only in the LHbLO that had a very specific innervation from the entopeduncular nucleus [66], a pivotal structure of the basal ganglia. Because the habenula-entopeduncular nucleus is often related to the aversive effect after rewarding [72], is reasonable to suppose that the decrease in astroglia in this nucleus is a neuroplasticity response to a synaptic modulation that occurs during the learning process. Thus, the activation of dopamine receptors expressed in astrocytes could be able to increase (DR1) or decrease (DR2) activation [73]. Hence, considering the role of dopamine in avoidance behavior [74] and the pivotal regulatory role of habenula in the dopaminergic system, is possible that the decrease observed in astroglial immunoreactivity is an example of an adaptive response from the aversive learning paradigm that results in a good performers in the avoidance protocol.

#### Limitations and Future Perspectives

An important limitation of this work is that we did not aim to identify the great number of neuronal populations activated, inhibited or affected by stress or learning performance. Rather, we aimed to demonstrate that habenular complex subnuclei show different neuronal and astroglial patterns in the paradigm of learning performance. Another important limitation of our study is that only male rats were included. It is well-known that females have a distinct response to aversive paradigms, and therefore, it is of the utmost importance that future works investigate sex differences regarding aversive learning and the regarding mechanisms and pathways involved. Future investigations should aim to understand the chronic and acute effects of avoidance and freezing behavior in the MHb-IPN pathway, focusing on the glutamatergic-cholinergic pathway and the habenular noradrenergic response. It would also be interesting to further understand the maladaptive neuroplasticity induced by chronic stress in the MHbI, LHbMPc and LHbLB. More specific work attempting to identify the cellular populations and connections involved in this paradigm is necessary to better comprehend the role of the habenular complex in stress and aversive learning.

## 4. Materials and Methods

### 4.1. Experimental Design

The experimental design is illustrated in Figure 5. After habituation to the animal facility, Wistar rats were submitted to eight daily training sessions in the two-way shuttle box and subsequently divided into good and poor performers according to the avoidance response exhibited during the sessions. The control animals were exposed to the two-way shuttle box, but the footshock was turned off. On the eighth day, 90 min after the end of the testing session, animals were anesthetized, perfused and the brains were collected for future analysis. Brains were sliced in a freezing microtome and processed for determination of histological landmarks (Nissl-stained), neuronal activation (FOS) and astrocyte density (GFAP) in order to compare control, good and poor performers. The immunoreactivity was evaluated in habenula’s medial and lateral complexes.

### 4.2. Animals

Male Wistar rats weighing 200–300 g were housed in polypropylene cages (40 × 34 × 17 cm) in groups of three. The room temperature was maintained at 24 °C ± 1 °C under a 12:12 dark/light cycle; wood shavings and free access to food and water were provided throughout the experiment. Animals were maintained for 7 days before experiments for habituation. All animal experiments were conducted and reported in accordance with the ARRIVE guidelines (http://www.nc3rs.org.uk/arrive-guidelines), accessed on 1 March 2022. The protocols used in this project were approved by the Ethics Committee on the Use of Animals at Hospital Sirio-Libanes (CEUA #2013/12).

### 4.3. Two-Way Active Avoidance Test

The paradigm was performed as previously described [75]. In brief, animals were submitted to 25 min training sessions once a day for 8 consecutive days in a two-way shuttle box (Insight Equipment, Ribeirao Preto, Brazil). Shuttling between compartments delayed the delivery of scrambled footshock unconditioned stimulus (US, 0.6 mA; 0.5 s) by 30 s. In the absence of shuttling, US delivery occurred every 5 s. The response-to-stimulus interval (R–S) shuttles comprised avoidance responses, and the stimulus-to-stimulus interval (S–S) shuttles comprised escape responses. All shuttles produced 0.3 s feedback stimuli (house light blink). Controls animals were exposed to the two-way shuttle box, but the footshock was turned off (i.e., without behavior contingency). Animals performing more than 20 avoidance responses in a session for 2 consecutive days were considered good performers, while animals that did not achieve this number were considered poor performers [7,75,76]. Freezing, defined as the absence of movement except that required for breathing [77], was assessed during the first 2 min of the tests [74,76].

### 4.4. Perfusion

Ninety minutes after the end of the last session, all groups of animals, including control, good and poor performers, were deeply anesthetized with thiopental (40 mg/kg) and morphine sulphate (10 mg/mL) and transcardiacally perfused with a solution of 0.9% phosphate-buffered solution followed by 4.0% paraformaldehyde in 0.1 M phosphate buffer, using a peristaltic pump (Cole Parmer, Vernon Hills, IL, USA). Brains were removed, placed in paraformaldehyde for 3 h and then transferred to a 30% sucrose/0.1 M phosphate buffer at 4 °C.

### 4.5. Microtomy

Frozen whole brain coronal sections (30 µm thick) were sliced with a sliding microtome (Leica SM 2000 R; Biosystems, Nussloch, Germany), collected and stored free-floating in PB 0.01 M for immunohistochemical assay.

### 4.6. Immunohistochemistry

Brain sections were processed overnight with anti-FOS antiserum raised in rabbit (Ab-5, Calbiochem, lot-D07099, Darmstadt, Germany; dilution 1:20,000) or mouse anti-GFAP (Sigma-Aldrich, Burlington, USA, catalogue# C9205, dilution: 1:1000) followed by incubation in appropriate biotinylated secondary antibody (Jackson ImmunoResearch, Ely, UK; dilution: 1:200) at room temperature for 2 h and avidin-biotin-peroxidase complex (ABC, Vector Laboratories, Newark, USA). The antibody complex was visualized via exposure to a chromogen solution containing 0.02% 3,3′-diaminobenzidine tetrahydrochloride (DAB; Sigma) in 0.05 M Tris-buffer (pH 7.6) and 0.01% hydrogen peroxide. Extensive washing in PBS buffer (pH 7.4) halted the DAB reaction. Additionally, a separate slide was used for negative reagent controls. The immunohistochemistry process was based on previous published guidelines of recommendations [78,79]. It includes the choice of the antibody titration to optimize the concentration and have the best measure of expression levels, using a commercially prepared and validated kit for performing the immunohistochemistry, including negative controls to identify any background staining. Additionally, the antibody cross reactivity was also evaluated using the percentage homology of the antibody immunogen to other similar proteins. We have used this same process through the years with the same care [7,11,74,80,81,82]. Sections were mounted on gelatin-coated slides, dehydrated and coverslipped with DPX (Sigma).

### 4.7. Quantification

Images were captured using a light microscope (E1000, Nikon, NY, USA) and quantification was performed using ImageJ software (News Version 1.44b National Institutes of Health, MD, USA; http://rsbweb.nih.gov/ij/, accessed on December 2022). An observed blinded-to-group allocation analyzed the FOS immunoreactivity (FOS-IR) and GFAP immunoreactivity (GFAP-IR) of the habenula subnuclei. FOS-IR was analyzed with stereology in 3–5 coronal sections per animal. GFAP-IR was analyzed using the threshold plug-in available in ImageJ software. For that, the background was subtracted from the images and the threshold was highlighted to type a known range of pixel intensities, then the particles were analyzed and the total of all particles was provided by the ImageJ. Additionally, the delineation of the habenula was performed and the corresponding area measurements were performed. The results were normalized by defining the control group as 100%. Border delineation, cell counting and area measurements were conducted with Image-J software (Version 1.44b). Adjoining Nissl-stained sections provided histological landmarks for the accurate identification and delineation of habenula subnuclei [31]. Figure 6 shows the subdivisions of the habenula. The habenula was first divided into medial (MHb) and lateral (LHb) regions. The medial complex (MHb) was divided into superior (MHbS), inferior (MHbI), central (MHbC) and lateral (MHbL) regions. The lateral complex (LHb) was subdivided into lateral (LHbL) and medial (LHbM) subdivisions and further parcellated into smaller subnuclei. The LHbL was divided into magnocellular (LHbLMc), oval (LHbLO), basal (LHbLB), parvocellular (LHbLPc) and marginal (LHbLMg) parts. The LHbM was parcellated into superior (LHbMS), parvocellular (LHbMPc) and central (LHbMC) parts. The corresponding Bregmas were from −3.00 mm to −4.36 mm, according to the Paxinos and Watson Atlas [83].

### 4.8. Statistical Analyses

Data are presented as the mean ± standard error of the mean (SEM). The sample size calculation was performed as previously described [84]. Statistical analyses were conducted with GraphPad Prism 9.0 software (GraphPad Software Inc.; San Diego, CA, USA). The normal distribution of the samples was confirmed using the Shapiro–Wilk test. The results of the two-way shuttle boxes test were analyzed using two-way repeated measures analysis of variance (ANOVA) considering Factor 1 Group, Factor 2 Session, followed by Bonferroni post hoc test. The immunohistochemistry assay results were normalized by defining the control group as 100% and analyzed using one-way ANOVA followed by Tukey post hoc test. For all tests, *p* < 0.05 was considered statistically significant.

## 5. Conclusions

The data presented here suggest that specific subdivisions of the MHb and LHb have different activation patterns and astroglial immunoreactivity in animals presenting good and poor avoidance behavior. We hope that this detailed evaluation will provide the basis for further studies to better comprehend the individualized signature and connectivity of each habenular subdivision.

## Figures and Tables

**Figure 1 ijms-24-10693-f001:**
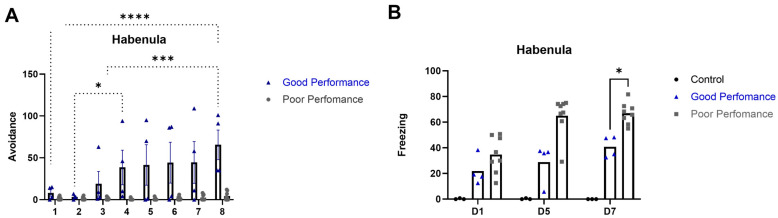
(**A**). Number of avoidance responses exhibited by good and poor performers animals throughout the training sessions. *, *p* < 0.05 good vs. poor performers; *** *p* < 0.001 good vs. poor performers; **** *p* < 0.0001 good vs. poor performers. (**B**). Percentage of time spent freezing in sessions 1, 5 and 7 considering control, poor and good performers. Data are presented as the mean ± standard error of the mean (SEM). *, *p* < 0.001 vs. good performance and control animals considering freezing behavior. Good performance (*n* = 8), poor performance (*n* = 4) and control (*n* = 3) animals.

**Figure 2 ijms-24-10693-f002:**
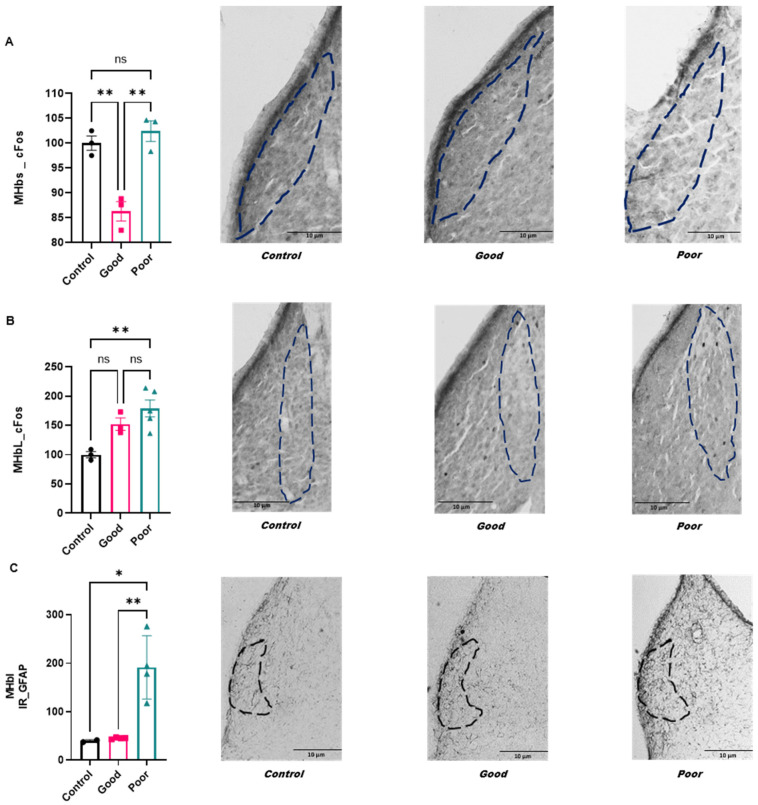
(**A**). FOS-IR in MHbS comparing control (*n* = 3), good (*n* = 3) and poor performance animals (*n* = 3). Data are presented as the mean ± standard error of the mean (SEM). (**B**). FOS-IR in MHbL of control *(n* = 3), good (*n* = 3) and poor performance animals (*n* = 5). (**C**). Quantification of GFAP-IR in MHbI comparing control (*n* = 3), good (*n* = 4) and poor performance animals (*n* = 4). The value corresponding to each animal was average, considering the right and left sides. The results were normalized by defining the control group as 100%. Data are presented as the mean ± standard error of the mean (SEM). *, *p* < 0.01; **, *p* < 0.01. Dotted lines of different colors correspond to individual animals.

**Figure 3 ijms-24-10693-f003:**
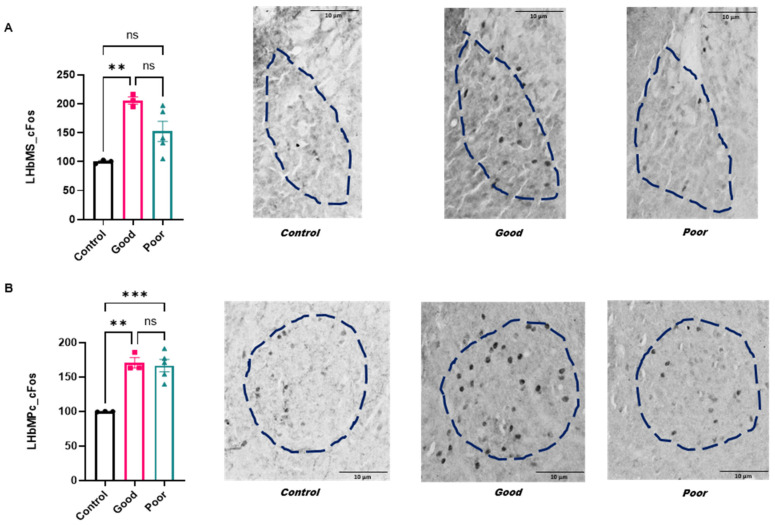
(**A**). FOS-IR in LHbMs. (**B**). FOS-IR in LHbMPc of control *(n* = 3), good (*n* = 3) and poor performance animals (*n* = 5). The value corresponding to each animal considers the right and left sides. The results were normalized by defining the control group as 100%. Data are presented as the mean ± standard error of the mean (SEM). ** *p* < 0.01 vs. control group; ***, *p* < 0.001 vs. poor performers. Dotted lines of different colors correspond to individual animals.

**Figure 4 ijms-24-10693-f004:**
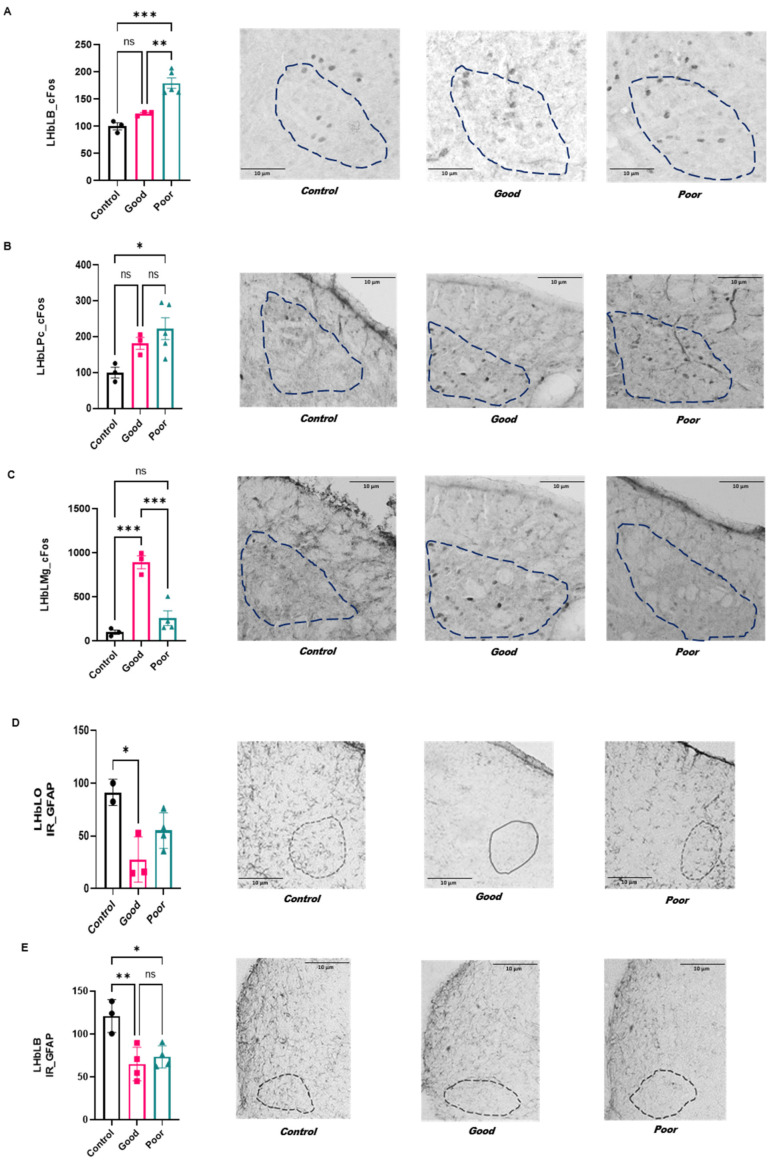
(**A**). FOS-IR in LHbLB, (**B**). in LHbLPc, (**C**). in LHbLMg of control (*n* = 3), good (*n* = 3) and poor performance animals (*n* = 4). (**D**). Quantification of GFAP-IR in LHbLB, (**E**). in LHbLO compared with control (*n* = 3), good (*n* = 4) and poor performance animals (*n* = 4). The value corresponding to each animal was average, considering the right and left sides. The results were normalized by defining the control group as 100%. Data are presented as the mean ± standard error of the mean (SEM). ** *p* < 0.01 vs. control group; *, *p* < 0.05; **, *p* < 0.01; ***, *p* < 0.001. Dotted lines of different colors correspond to individual animals.

**Figure 5 ijms-24-10693-f005:**
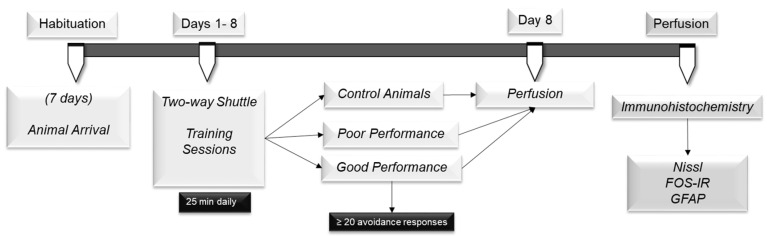
Experimental design of the study. After habituation to the animal facility, animals were submitted to eight daily training sessions in the two-way shuttle box. On the eighth day, 90 min after the end of the session, animals were anesthetized and perfused and the brains were processed for determination of histological landmarks (Nissl-stained), neuronal activation (FOS) and astrocyte density (GFAP).

**Figure 6 ijms-24-10693-f006:**
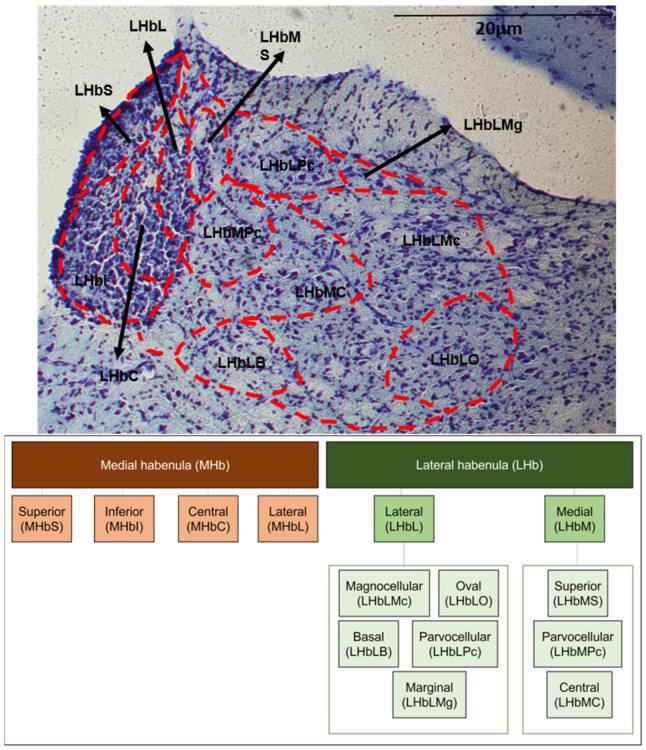
The subdivisions of the habenula in Nissl staining and schematic. Habenula was first divided into medial (MHb) and lateral (LHb) complexes. The medial complex (MHb) was divided into superior (MHbS), inferior (MHbI), central (MHbC) and lateral (MHbL) regions. The lateral complex (LHb) was subdivided into lateral (LHbL) and medial (LHbM) regions. The LHbL was further parcellated into magnocellular (LHbLMc), oval (LHbLO), basal (LHbLB), parvocellular (LHbLPc) and marginal (LHbLMg) parts, while the LHbM was parcellated into superior (LHbMS), parvocellular (LHbMPc) and central (LHbMC) parts.

**Table 1 ijms-24-10693-t001:** Summary data showing behavioral, FOS -IR and GFAP-IR in good and poor performers.

	Good Performers	Poor Performers
Behavioral Data	Increased avoidance response	Decreased avoidance response
	Decreased freezing behavior	Increased freezing behavior
FOS-IR	Increased LHbLB	Increased LHbLB
	Decreased MHbS	
GFAP-IR		Increased MHbI

## Data Availability

The data generated and analyzed during the current study are available from the corresponding author upon reasonable request.

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
