# Peer review of "Unravelling the Role of Habenula Subnuclei on Avoidance Response: Focus on Activation and Neuroinflammation"

_ijms, 2023, doi:10.3390/ijms241310693_

Round 1
Reviewer 1 Report
The manuscript by Geiza Fernanda Antunes and co-workers describes a study where the neuronal activity pattern and the astroglia morphology of the habenular subnuclei was tested in rats showing different pattern in avoidance behavior.
The study must be improved in order to increase the scientific value of the paper.
Major critical points:
1. The ethical permission the Authors refer to is almost 10 years old. Authors may add how long is/was this permission valid.
2. Quality of histological images regarding c-Fos. The FOS protein is the subunit of the AP1 transcription factor that is translocated into the nucleus if the gene transcription is increased. Therefore, the FOS labeling must show a nuclear immunosignal. This is not visible on any images related to this staining.
3. It is unclear what exactly do the graphs in Figs 3-8 show. What do the numbers indicated on the vertical exis refer to? What was measured? Is this optical density or cell counts? How was this corrected for the area and/or background signal? This should be described in the methods section, because there are many ways of morphometrical assessments that one can perform using ImageJ.
4. The discussion must be rewritten because it lists many anatomical and neurochemical facts, but no efforts were done in this work to identify what cells did show higher or lower neuronal activity regarding FOS for example by double labelings. Authors write many suggestive statements that remain possibilities only because they do not provide experimental evidence that what cell types were activated or showed less activity.
5. How do Authors interpret the findings regarding the astroglia? These results were not discussed.
6. Nomenclature of the examined protein FOS. Authors refer to c-Fos, sometimes to cFos, or c-fos. The protein they examined is officially called FOS.
7. Antibody controls missing.
8. GFAP catalog number missing.
9. It is unclear why an acute activation marker was examined if the rats were tested for 8 days. DeltaFOSB/FOSB may have been the right choice.
Minor points:
1. The scale bars are missing in the histological figures.
2. Fig1B suggest that control rats got exposed to training sessions, but in the text it is stated that they did not suffer foot shocks. This should be corrected. Also in Fig 1B Nissl is mis-spelled ("Nissil").
3. The experimental design paragraph may be moved forward to the first part of the Methods section.
4. Authors say they perfused that rats with PFA. Usually this is done after a perfusion with saline or PBS.
Authors may check the manuscript for grammatical and typographical errors. some examples:
ln 10 "GFPA"
ln 58 have
ln 76 "specific pattern of activation and inflammatory"- This requires rephrasing.
In the graphs of Figures 3-8 "Figura"
ln 301 receives
ln, 383. It is unclear what does "Taken together..." refer to in this context.
Reviewer 2 Report
This study by Antunes et al. is designed to understand the role of the habenular complex subnuclei on avoidance responses. They reported that rats with higher avoidance activity when compared to rats with lower avoidance activity showed distinct phenotypes of c-fos activity and astroglial reactivity in specific habenular nuclei. Despite the interesting topic of this study, the role of habenula on avoidance responses is not a new topic. In addition, there are some issues how sample sizes were derived and statistical approach for support data comparisons. The discussion clearly suffers from objectivity. Per my comment above – the manuscript needs revision.
There are some concerns:
(1) Statistical tests: Three parametric tests are introduced as part of the plan of data analysis. It is assumed that all data follows a normal distribution? Did the authors check the assumptions required for parametric analysis (i.e., level of measurement, normality, homogeneity of variances)? How sample size is pre-estimated? The authors stated: “Pearson correlation test was performed to compare two quantitative variables, where r>0.75 was strongly correlate”. Throughout the manuscript I am not able to identify the application of such analysis. Please ensure that the appropriate descriptive statistics are used for the type of data under study.
(2) To properly interpret the study, the reader must be able to evaluate potential bias due to lost, missing or excluded data. Please report if any data were missing or lost for any reason. The authors do not state a reason for omitting females in this study.
(3) (Figure 1B): Regarding the experimental design illustration. If we look carefully is not clear to what protocol the control animals were exposed (only shuttle box without behavior contingency, or shuttle box with behavior contingency) and if they were perfused after exposition. The authors should clarify this issue.
(4) (Figure 2): Shuttle box responses. For each animal, please represent the individual responses per training session. The same can be extended to panel B.
(5) The authors showed that good performance rats increased the number of avoidance responses and time spent in freezing after learning sessions. If the animals learn and improve their accuracy to avoid the aversive stimuli. It is not supposed to see a reduction of freezing behavior? The authors should comment this result.
(6) The authors should keep figures concise. There are too many figures and some panels will benefit by incorporation in the same figure. This is the case of figures 3 and 4; 6, 7 and 8. In addition, less important data can be transferred to supplementary data such as the tables 1 and 2. The authors should do a better work in the legends of plots axis to help the reader to understand what is represented. Moreover, the manuscript would benefit if the authors included a final table summarizing the changes per behavior response, brain nuclei, and c-fos/GFAP.
(7) Morphological and structural profiles of habenular complex. Like the authors clearly stated the habenular complex is not a homogeneous brain region. Regarding the immunohistochemical characterization. Did the authors used only a single coronal slice or the average of multiple slices per animal? The figures legends would benefit with the inclusion of the location of the coronal slice. From the methods section, it is not clear the methodology applied using the Fiji software to quantify tissue changes. The authors should clarify this issue. Moreover, the authors normalized the immunohistochemical data of the control group to be represented as 100% of the effect. From my point of view this is not the most appropriate way to represent the immunohistochemical data from independent data. The authors should justify their option for use this representation of the data.
(8) The authors need to improve manuscript writing, especially improving the introduction in a more objective way and the discussion to support their main achievements. In the discussion section, I found the paper hard to read. Much emphasis is given to other research reports, instead to discuss the present data and their relevance in a focuses manner. In the conclusion paragraph it is missing a statement about the limitations of this study, future questions to be solved or potential translations aspects.
(9) There are references with the publication date in Portuguese. Please format the references using the layout of the journal.
I did not identify any significant problems.
Round 2
Reviewer 1 Report
Authors addressed some of the concerns of this reviewer. The following points remain to be treated:
Omission control of antibodies does not exclude the cross reactivity with other targets. How do Authors know that their antibodies were really specific?
The assessment of FOS-ir is clearer now. But, Authors may explain more accurately how did they evaluate the GFAP labeling. Was this the ratio of surface area that the GFAP immunoreactivity covers in a field of interest? How was this measured? Did Authors perform a skeletonization after thresholding and they measured the area in a binary image? This should be explained in a way that one can replicate the study.
There is some improvement in the histological images. However, they are not always representative to the results indicated in the corresponding graphs. For example, in Fig 4B, in the marked area hardly any FOS-ir nuclei can be recognized, but the graph shows 100 cells per mm2.
Authors added scale bars, but they don't indicate its length in micrometers. Instead, the magnification is indicated, but using a 100x or a 50x objective is not the same as the final magnification. On one hand, the camera's lens further magnifies the picture, and the magnification of the objective lens multiplied by the magnification of the ocular/camera lens provides the final magnification. Also, the digital image processing allows arbitrary digital zoom that changes the magnification value, again. Therefore, it would be better to add the lengh of the indicated bars in micrometers, and in the legends, if Authors wish, they may indicate the magnification, but it should be the true, final magnification and not the objective lens magnification value only.
For some reason in Fig. 6E even 2 bars per image appear. This is superfluous.
There is no considerable improvement in the discussion section. Author left the superfluous anatomical descriptions about connectivity of these centers in the text that do not help the reader to understand the findings (examples: pg 13, 2nd and 3rd paragraph, pg 14 2nd paragraph). These parts may be moved into a heading about the anatomy of habenular complex in a review paper on habenula. But, in this original research paper these parts do not help the reader to interpret the findings. Also Fig 7 shows a summary of the connectivity that was not studied in this work, and it is like a typical review paper figure, except if a paper is about the mapping of connectivity. But, this is not the case now.
The second paragraph in the Discussion is also too lengthy, it could be radically shortened.
Instead of the long anatomical descriptions of connectivity Authors may discuss and cite the findings of other researchers performing FOS labeling in the habenula and compare their results with the current ones. Some relevant papers as examples that Authors may use to compare the present findings with: PMID: 8137158, 35993349, 35592695, 28622391, 27785462). Authors may check for similar papers studying GFAP also.
This reviewer appreciates the limitation part added to the R1 version. However, the first 9 lines repeat information from earlier parts. This can be deleted.
In the figure legends authors use the term "hemisphere". With regard to the habenula it is at least questionable if this is the right term, because the epithalamic habenular area is not part of the telencephalon, that would have hemispheres in the brain, (besides the cerebellum, of course). It is unusual to apply the term hemisphere in the diencephalon.
Still some grammatical error can be found. Example: pg 14. 3rd par. line 2: "have"
Reviewer 2 Report
The authors have done an adequate job to most concerns. However, there is an issue with the statistical methodology applied to compare the experimental data in the new Figure 3. The authors claim to have used a two-way ANOVA (which implies an interaction between two factors; exp. groups vs. testing sessions), this information is not correctly reported in the results section. In addition, it remains unclear which specific comparisons were performed. The authors should clarify these points. Overall, I have no further comments.
Round 3
Reviewer 1 Report
Authors answered the comments by this reviewer this time adequately.
One final remark is that Authors answered the question about antibody specificity excellently, but this does not appear in the revised manuscript. Therefore, this reviewer would suggest to add the citations and at least a sentence or a short paragraph about the lack of cross reactivity with closely related proteins to the Methods part about immunolabeling, and the citation(s) supporting this statement should appear also there in the manuscript, and not only in the response letter.
No other issues came up when reading the revised (R2) version.
Pg 10. last., highlighted sentence: Space missing before sentence, and the term "reaction" is somewhat confusing in this context. I would rather say either "astroglial activation to stress" or "increased/altered GFAP immunoreactivity upon stress exposure". (In one sense, "GFAP reaction" is the staining method to show astrocytes as GFAP is their marker. But, Authors want to say here that these cells respond to stress that ultimately alters the astrocyte morphology and the pattern of GFAP labeling.)
